# Planar EPID-Based Dosimetry for SRS and SRT Patient-Specific QA

**DOI:** 10.3390/life11111159

**Published:** 2021-10-30

**Authors:** Sangutid Thongsawad, Tadchapong Chanton, Nipon Saiyo, Nuntawat Udee

**Affiliations:** 1Faculty of Medicine and Public Health, HRH Princess Chulabhorn College of Medical Science, Chulabhorn Royal Academy, Bangkok 10210, Thailand; 2Department of Radiation Oncology, Chulabhorn Hospital, Chulabhorn Royal Academy, Bangkok 10210, Thailand; nipon.sai@pccms.ac.th; 3Department of Radiological Technology, Faculty of Allied Health Sciences, Naresuan University, Phitsanulok 65000, Thailand; waterboy11323@hotmail.com (T.C.); nuntawatu@nu.ac.th (N.U.); 4Faculty of Health Science Technology, HRH Princess Chulabhorn College of Medical Science, Chulabhorn Royal Academy, Bangkok 10210, Thailand

**Keywords:** SRS patient-specific QA, EPID-based dosimetry, FFF beam verification

## Abstract

The study’s purpose was to develop and validate Electronic Portal Imaging Device (EPID)-based dosimetry for Stereotactic Radiosurgery (SRS) and Stereotactic Radiation Therapy (SRT) patient-specific Quality Assurance (QA). The co-operation between extended Source-to-Imager Distance (SID) to reduce the saturation effect and simplify the EPID-based dosimetry model was used to perform patient-specific QA in SRS and SRT plans. The four parameters were included for converting the image to dose at depth 10 cm; dose-response linearity with MU, beam profile correction, collimator scatter and water kernel. The model accuracy was validated with 10 SRS/SRT plans. The traditional diode arrays with MapCHECK were also used to perform patient-specific QA for assuring model accuracy. The 150 cm-SID was found a possibility to reduce the saturation effect. The result of model accuracy was found good agreement between our EPID-based dosimetry and TPS calculation with GPR more than 98% for gamma criteria of 3%/3 mm, more than 95% for gamma criteria of 2%/2 mm, and the results related to the measurement with MapCHECK. This study demonstrated the method to perform SRT and SRT patient-specific QA using EPID-based dosimetry in the FFF beam by co-operating between the extended SID that can reduce the saturation effect and estimate the planar dose distribution with the in-house model.

## 1. Introduction

In radiation therapy, advanced beam delivery techniques such as Intensity Modulated Radiation Therapy (IMRT) are developed to increase the radiation dose at a tumor and reduce radiation dose at normal organs simultaneously [1,2]. The comprehensive step, namely patient-specific Quality Assurance (QA), was required for the IMRT process to assure the dose agreement between treatment planning calculation and beam delivery. Many QA tools were used to measure the radiation for patient-specific QA, such as detector arrays (Delta4, Matrixxx, Octavious, etc.), Electronic Portal Imaging Device (EPID), film, gel dosimetry, and log file-based systems. Due to the small field used in Stereotactic Radiosurgery (SRS) and Stereotactic Radiation Therapy (SRT), the patient-specific QA requires a high-resolution detector to catch up the error. The conventional detector arrays with low resolution can significantly affect the QA with false-positive results [3]. The high-resolution QA tools such as film dosimetry, gel dosimetry, and film-class resolution digital detector were widely used for SRS patient-specific QA. However, the practical drawbacks of each QA tool have been reported, such as time-consuming film and gel [4,5] and the time required for setup and alignment of detector arrays [6]. EPID-based dosimetry is another choice for SRS patient-specific QA that is less time-consuming and does not require setup and tool alignment. Many researchers developed the algorithm for EPID-based dosimetry with different approaches [7,8,9,10,11]. However, the saturation effect has been reported [12,13,14,15] according to high dose rate measurement in flattening-filter-free (FFF) beams. FFF beams are generally used for treatment in cases of SRT, SRS, and Stereotactic Body Radiation Therapy (SBRT) which was required high dose rates to reduce the treatment time [16,17,18]. The saturation effect can be solved by providing the new Image Acquisition Systems (IAS) from the vendor to catch up the high dose rate [19], and many studies were developed the method to reduce the saturation effect. Tyner et al. [20] demonstrated the method to reduce the saturation effect by placing 1 cm of water-equivalent plastic on the EPID surface. Nicoli et al. [21] extended Source-to-Imager Distance (SID), and they found SID up to 150 cm was suitable to reduce saturation effect in FFF beams. Chuter et al. [22] adopted the extended SID method with 160 cm to reduce the saturation effect for a dose rate less than 800 MU/min.

The purpose of this study was to develop and validate EPID-based dosimetry for SRS and SRT patient-specific QA. Four parameters were included for converting the image to a planar dose at depth 10 cm; dose-response linearity with MU, beam profile correction, collimator scatter and water kernel.

## 2. Material and Method

TrueBeam LINACs (Varian Medical Systems, Palo Alto, CA, USA) and the aS 1000 EPID detector systems with integrated mode using Image Acquisition Systems version 3 (IAS-3) were used in this study. The radiation treatment of SRS and SRT was delivered with a 6 MV FFF beam. Eclipse TPS version 13.6 (Varian Medical Systems, Palo Alto, CA, USA) was used to calculate dose distribution. MATLAB software version 2019b (The Mathworks, Inc, Natick, MA, USA) was used to manipulate the images in the EPID-based dosimetry model.

Figure 1 shows a flowchart diagram of this study. The appropriated SID to reduce saturation effect was investigated, and then the EPID-based dosimetry model for FFF beam was developed. The accuracy of our model (the co-operation between extended SID and EPID-based model) was validated by comparing between EPID-based dosimetry measurements and planned dose calculation in clinical plans. In addition, the traditional measurements with diode arrays were used to confirm the accuracy of our EPID-based dosimetry model by performing patient-specific QA in the same clinical plans.

### 2.1. Appropriate SID for Reducing Saturation Effect

In this study, the appropriate SID that can reduce the saturation effect was found by comparing the radiation measurement in different SIDs from 120 to 180 cm, and the measurement was performed with different dose rates (400, 600, 800, 1000, 1200, 1400 MU/min). This experiment’s hypothesis is the radiation signal should similar when delivering the same MU in different dose rates. The standard deviation (SD) of EPID signals was used to determine the EPID saturation effect in different SIDs.

### 2.2. EPID-Based Dosimetry Model

Before applying the EPID-based dosimetry model to the measurement, basic image calibration was performed according to vendor recommendations [19], including darkfield, flood field, and dose normalization. The EPID-based dosimetry model was designed by converting the EPID images (*EPID_x,y_*) to absorbed dose in the water at a depth of 10 cm (*D*) with four parameters as shown in Equation (1).
(1)$$D=(EPID_{x,y}\;\times \;f_{dose}\;\times \;M_{FFF}^{2D}\;\times \;S_{fs})\;{⊗}^{-1}\;K_{water}$$

#### 2.2.1. Dose Linearity Calibration

The first step, the relation between Calibration Units (CU) and absorbed dose at a depth of 10 cm in water (*f_dose_*), was determined for scaling CU to absorbed dose in Gy, and it can be explained as Equation (2).
*f_dose_* = *A* × CU + *B*(2)

*A* is the first constant parameter for linearity function, and *B* is the second constant parameter for a linear function.

To find the function of linearity dose calibration, the absorbed dose in water was measured using FC65-G cylindrical ionization chamber (IBA Dosimetry GmbH, Schwarzenbruck, Germany) for various MU with the field size of 10 × 10 cm^2^ at a depth of 10 cm, which was related to radiation measurement from EPID in CU.

#### 2.2.2. 2-Dimensional Beam Profile Correction

The second step, the off-axis difference between EPID and water, was assessed using the ratio between water and EPID diagonal profile at a field size of 40 × 40 cm^2^. A polynomial fourth order function was determined to fit the beam profile correction curve, and then the function was extracted into 2-dimensional symmetry ($M_{FFF}^{2D}$), which was explained as Equation (3).
(3)$$M_{FFF}^{2D}=Cx^{4}\;+\;Dx^{3}\;+Ex^{2}\;+Fx+G$$

*x* is the off-axis distance in cm, whereas *C*, *D*, *E*, *F*, and *G* are the first, second, third, fourth, and fifth constant parameters.

#### 2.2.3. Collimator Scatters Correction

The third step, the influence of collimator scatter difference between EPID and water was calculated using the ratio of collimator scatter response between EPID and water. A polynomial third-order function was determined to fit the collimator scatter correction curve (*S_fs_*), which was explained as Equation (4).
*S_fs_* = *Hx*^3^ + *Ix*^2^ + *Jx* + *K*(4)

*x* is the equivalent square field size in cm × cm, whereas *H*, *I*, *J*, and *K* are the first, second, third, and fourth constant parameters.

To find the scatter correction, the collimator and phantom scatter were measured using CC01 and FC65-G cylindrical ionization chamber (IBA Dosimetry GmbH, Schwarzenbruck, Germany) at a depth of 10 cm with varying field size from 1 × 1 cm^2^ to 8 × 8 cm^2^, which were related to radiation measurement from EPID in CU. To improve the accuracy of scatter measurement by an ionization chamber, the collimator, and phantom scatter was conducted with the method of IAEA TRS 483 protocol [23].

#### 2.2.4. Water Kernel

The fourth step, a remaining error between our model corrected with previous corrections (*f_dose_*,$\;M_{FFF}^{2D}$, *S_fs_*) and dose distribution in water, was reduced using the water kernel (*K_water_*) parameter as described by King et al. [24] method. Briefly explained here, the optimization algorithm was used to fit the water kernel (*K_water_*) function by minimizing disagreement between the EPID-based dosimetry profile and the ionization chamber profile. This model was predominantly used for SRS/SRT patient-specific QA, therefore; the small field profile with field sizes of 2 × 2, 3 × 3, 5 × 5, and 8 × 8 cm^2^ was collected for the optimization in this study. Basically, an exponential function was used to fit the water kernel (*K_water_*) function, which was explained as Equation (5). The water kernel function was extracted into 2-dimensional symmetry to operate with EPID images.
(5)$$K_{water}=e^{-\left(a_{1}r\right)}+a_{2}e^{-\left(a_{3}r\right)}+a_{4}e^{-\left(a_{5}r\right)}$$
where *r* is the distance from beam central-axis and *a*_1_, *a*_2_, *a*_3_, *a*_4_, *a*_5_ are the first, second, third, fourth, and fifth constant parameter, respectively.

### 2.3. Model Validation

Figure 2 shows the process of model validation. To assess the model accuracy, 10 SRS/SRT plans in brain lesions were randomly conducted for the patient-specific QA measurements. Table 1 shows plan information for this study. The information of collimator field size and PTV size was also addressed, as shown in Table 2. Dose agreement between EPID-based dosimetry and plane dose calculation from TPS was determined using gamma analysis with gamma criteria of 3%/3 mm, 2%/2 mm, and the cut-off threshold at 10%.

In our department, MapCHECK diode arrays (Sun Nuclear Corporation, Melbourne, FL, USA) were previously used as a QA tool; hence, dose agreement between MapCHECK measurement and plane dose calculation from TPS was also determined to confirm the result of model validation. According to the low resolution of Mapcheck detector (7.07 mm spacing for inner area), it is not enough accuracy for measurement of the small field in SRS and SRT. Therefore, the resolution of the MapCheck detector was increased by merging two measurement sets with SNC Patient software. The first measurement was performed by aligning beam isocenters to the center of MapCheck, and the second measurement was performed by aligning beam isocenter to the phantom offset at 5 mm superior with couch shift. This method can increase the detector resolution to 3.54 mm spacing for the inner area.

For dose calculation from TPS, plan data was transferred to slab water phantom (size of Height × Width × Long: 40 × 30 × 40 cm) by resetting the gantry to 0 degrees. Then the dose distribution was re-calculated using AAA (Analytical Anisotropic Algorithm) of Eclipse TPS. The dicom format of the single plane dose was exported to MATLAB software to verify the dose agreement between our EPID-based dosimetry and plane dose calculation from TPS. For EPID-based dosimetry, the image was quired with integrated mode, and then images were calculated to plane dose at a depth of 10 cm with our model.

## 3. Results

### 3.1. Appropriate SID for Reducing Saturation Effect

Table 3 shows the EPID signal in CU with varying SID and dose rates. The SD of the EPID signal was used to determine the EPID saturation effect in different SIDs. The maximum SD was found at SID of 120 cm with 3.45 SD in a measurement dose rate from 400 to 1400 MU/min. The minimum SD was found at SID of 180 cm with 0.64 SD in a measurement dose rate from 400 to 1400 MU/min. However, the increasing of SID may produce the scattered radiation with increasing the noise to EPID measurement. Hence, the SID of 150 cm was selected for patient-specific QA to compromise between reducing the saturation effect and scattered radiation.

### 3.2. EPID-Based Dosimetry Model

Linearity function (*f_dose_*) for EPID-based dosimetry was found as *f_dose_* = 0.008 × CU + 0.011. Figure 3 shows the relationship between CU and absorbed dose (Gy) at a depth of 10 cm.

The polynomial function of one-dimensional beam profile correction was found as $1D\;\mathrm{profile}=3.9\times 10^{-4}x^{4}+1.7\times 10^{-2}x^{3}-0.29x^{2}+0.42x+100$. Then 1D profile was extracted to a 2D symmetry profile, as shown in Figure 4.

Collimator scatter correction (*S_fs_*) was found as *S_fs_* = −7.8 × 10^−5^*x*^4^ + 0.0022*x*^3^ − 0.023*x*^2^ + 0.11*x* + 0.75. Figure 5 shows collimator scatter correction (*S_fs_*) plotted between radiation field size and collimator scatter correction.

Water kernel function (*K**_water_*) was found as *K**_water_* = *e* ^(−25∗*r*)^ + (8 × 10^−4^) *e* ^(−1.5∗*r*)^ + (1.7 × 10^−5^) *e* ^(−0.22∗*r*)^. Where r is distance from the center. Figure 6 shows water kernel (*K**_water_*) curve for 6 X-FFF.

### 3.3. Model validation

Table 4 shows gamma passing rates (GPR) results of our EPID-based dosimetry and MapCHECK measurements. For EPID-based dosimetry at criteria of 3%/3 mm, lowest GPR was 98.99 ± 0.73% and highest GPR was 99.8 ± 0.39%. For EPID-based dosimetry at criteria of 2%/2 mm, lowest GPR was 95.33 ± 2.95% and highest GPR was 99.47 ± 3.12%. For MapCHECK at criteria of 3%, 3 mm, lowest GPR was 96.90 ± 1.57%, and highest GPR was 99.66 ± 2.1%. For MapCHECK of 2%, 2 mm, lowest GPR was 94.83 ± 3.03% and highest GPR was 98.61 ± 2.49%. Figure 7 shows example of planar dose distribution between EPID-based dosimetry and TPS dose calculation with gamma criteria 2%/2 mm.

## 4. Discussions

In this study, the saturation effect was reduced by using the extended SID method. Although SID increasing can decrease the saturation effect, the scatter radiation (noise) also increases when SID was increased [25]. Hence, the SID of 150 cm was selected as the appropriate SID for EPID measurement with a signal range between 497.18 CU (at dose rate 1400 MU/min) and 500.81 CU (at dose rate 400 MU/min), and the standard deviation (SD) of measurement was found at ±1.35 CU. The appropriated SID was found comparable to Pardo study [13], with the SID beyond 140 cm can reduce the saturation effect.

There are two approaches to reconstruct EPID-based dosimetry. The first approach, photon fluence was calculated to dose distribution at the EPID plane using a TPS or independent algorithm. Then the dose distribution at the EPID plane was compared to EPID measurement [26]. In the second approach, EPID images were calculated to the absorbed dose at the water. Then the dose estimation at the water was compared to TPS’s dose calculation [27,28]. The advantage of the second approach is the potential to directly verify the TPS algorithm’s accuracy [29]. Our EPID-based dosimetry model was also achieved as a second approach by reconstruction images to planar dose distribution in the water.

Our EPID-based dosimetry model has been published elsewhere [30] with less validated plans. Therefore, this study has the effort to implement the EPID-based dosimetry model for SRS and SRT plans, and the results demonstrated a good agreement between EPID-based dosimetry and traditional detector arrays measurement (MapCHECK).

MapCHECK measurement represents a previous patient-specific QA tool in our institute with low spatial resolution detectors (445 diodes in 22 × 22 cm^2^ of radiation area) compared to EPIDs with a high spatial resolution (1024 × 768 pixels in 40 × 30 cm^2^ of radiation area). Although the density of the detector was increased with merging between two measurement sets, the MapCheck detector resolution is still less than EPIDs. When the results of patient-specific QA using EPID-based dosimetry and MapCHECK measurements were observed, it was found that EPID-based dosimetry has better agreement than MapCHECK due to the influence of spatial resolution detectors as described by Benjamin et al. [31]. The result of our EPID-based dosimetry and MapCHECK was also compared to Miri study [12], and similar results were found.

The patient-specific QA procedure between EPID-based dosimetry and diode arrays was discussed here; patient-specific QA with diode arrays required setting a phantom and connecting the signal cable while patient-specific QA with EPID did not require them. However, EPID-based dosimetry required more frequency of dose calibration than diode arrays because the accumulated dose effect can reduce EPID signal sensitivity [32]. In this study, both QA tools were performed the dose calibration before measurement to eliminate the sensitivity degradation.

According to the image acquisition method, EPID-based dosimetry can be measured with two modes: integrated and cine mode. EPID captures a single image for integrated mode consisting of the average number of frames acquired during radiation delivery [26]. For cine mode, a sequence of multiple images is captured during radiation delivery instead of a single integrated image [33]. Since cine mode is synchronized to beam pulses, the frame acquisition rate depends on the dose rate [34]. Our study used the integrated mode to acquire the images due to a more negligible dose rate effect than the cine mode.

Backscatter form arm support was influenced to the accuracy of EPID-based dosimetry for amorphous silicon (a-Si) 1000 as described by [35]. To reduce the backscatter effect from arm support in this study, the Varian’s Preconfigured Portal Dosimetry Package (PDPC) was applied to EPID measurement with the 2D profile correction image [36].

The disadvantage of this study is the EPID-based dosimetry measurement does not include isocenter accuracy because EPID images were acquired by resetting the gantry at 0 degrees. However, the treatment isocenter accuracy was separately verified in the Winston-Lutz test.

In this study, the various clinical plan parameters such as dose prescription, arc number, target number, were included in the model validation for testing the model accuracy. However, the limitation of filed sizes from 2.6 to 6 cm^2^ was only validated in this study. Therefore, the field size < 2 cm^2^ should be concerned for this model. The GPR was found more than 98% for gamma criteria 3%/3 mm, and more than 95% for gamma criteria 2%/2 mm.

This study demonstrated the feasibility to use EPID-based dosimetry for SRS and SRT patient-specific QA. However, the commissioning and end-to-end test should be performed before clinical use with point dose measurement using ionization chamber and 2D dose distribution using detector arrays, film dosimetry, and gel dosimetry according to AAPM Medical Physics Practice Guideline 9.a [37] recommendation.

To maintain the EPID-based dosimetry accuracy, the periodic check with the other QA is recommended to track the radiation dose.

## 5. Conclusions

This study demonstrated the method to perform SRS and SRT patient-specific QA using EPID-based dosimetry in the FFF beam by co-operating between the extended SID method to reduce the saturation effect and estimate the planar dose distribution with in-house model. The possibility of SID that can reduce saturation effect was found at 150 cm with EPID measurement range between 497.18 to 500.81 CU, and SD of ±1.35 CU for dose rate range between 400 to 1400 MU/min. The model accuracy was validated in a range of field sizes from 2.6 to 6 cm^2^ and found good agreements between our model and TPS calculation with GPR more than 98% for gamma criteria 3%/3 mm, and GPR more than 95% for gamma criteria 2%/2 mm.

## Figures and Tables

**Figure 1 life-11-01159-f001:**
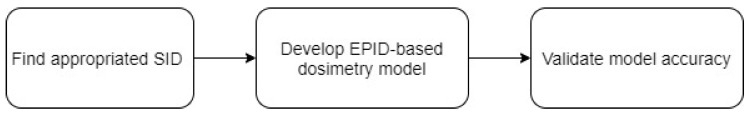
Flowchart diagram of this study.

**Figure 2 life-11-01159-f002:**
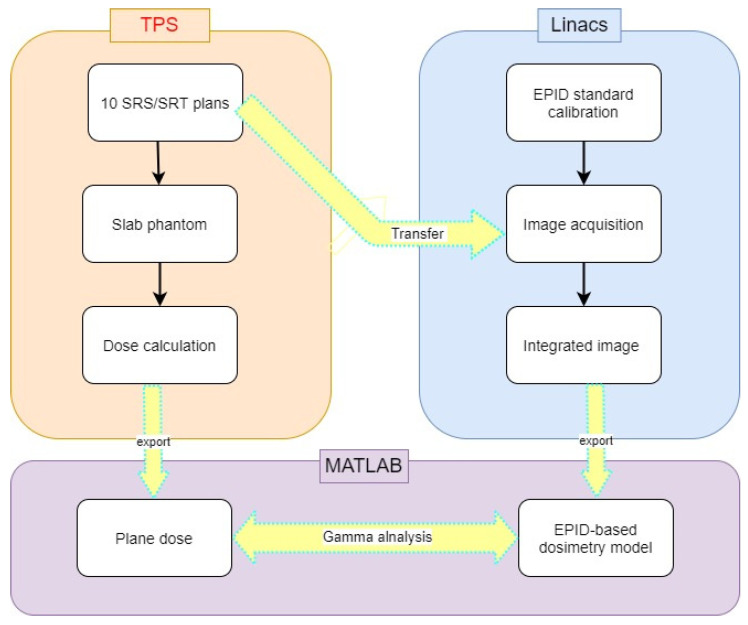
The process of model validation.

**Figure 3 life-11-01159-f003:**
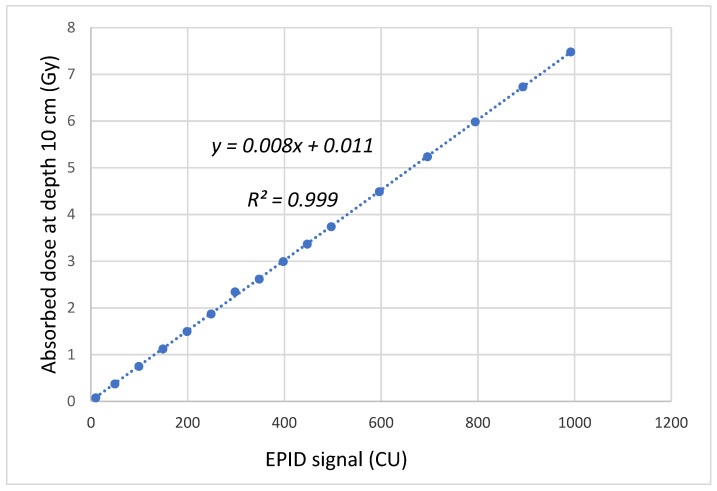
The relationship between CU and absorbed dose (Gy) at a depth of 10 cm.

**Figure 4 life-11-01159-f004:**
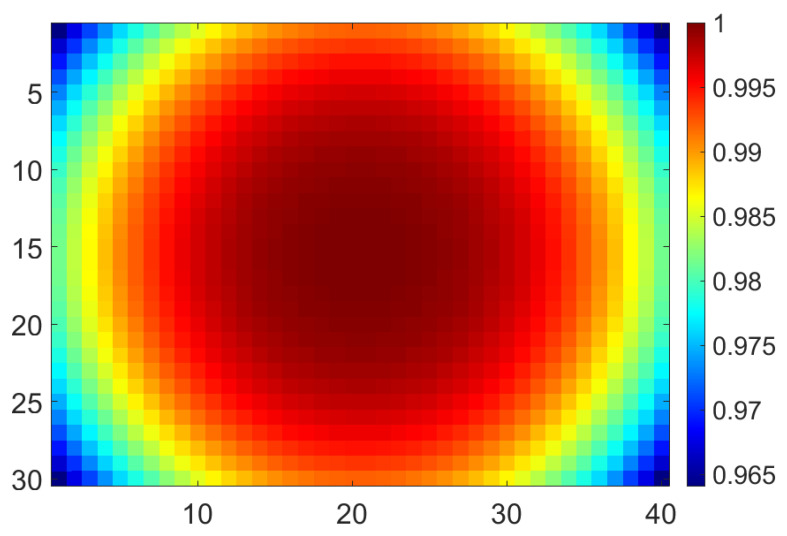
2D symmetry profile of this model.

**Figure 5 life-11-01159-f005:**
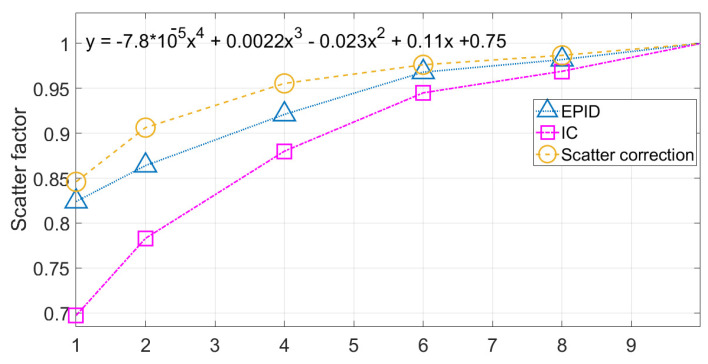
Collimator scatter correction (*S_fs_*), EPID collimator scatter (*S_c,p_ EPID*), and water collimator scatter (*S_c,p_ water*).

**Figure 6 life-11-01159-f006:**
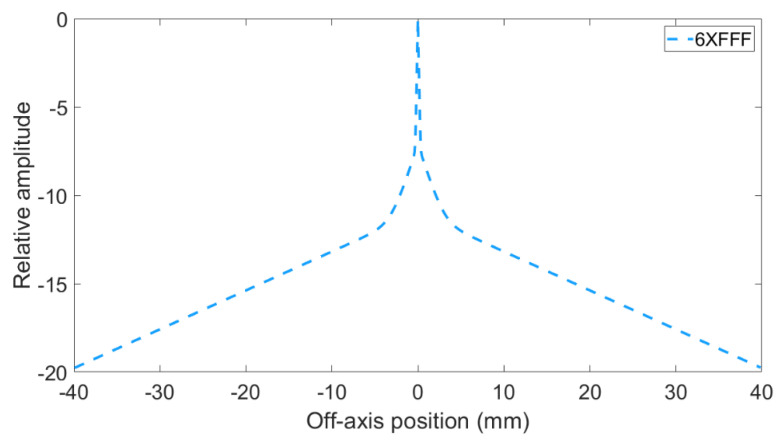
Water kernel (*K_water_*) of EPID for 6 X-FFF.

**Figure 7 life-11-01159-f007:**
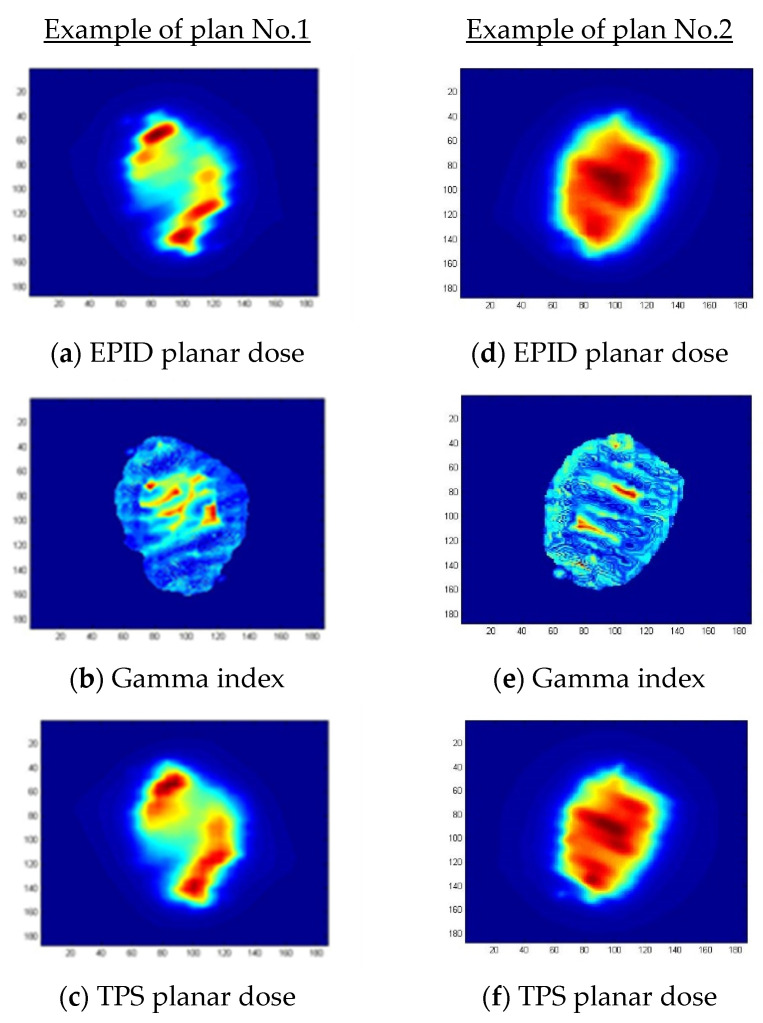
Example of planar dose distribution between EPID-based dosimetry and TPS dose calculation with gamma criteria 2%/2 mm, (**a**) EPID planar dosimetry of plan No.1, (**b**) gamma index of plan No.1, (**c**) TPS planar dose of plan No.1, (**d**) EPID planar dosimetry of plan No.2, (**e**) Gamma index of plan No.2, and (**f**) TPS planar dose of plan No.2.

**Table 1 life-11-01159-t001:** Information of model validation plans.

Plan Parameters	Information (n = Number of Plan)
Dose prescription	6 Gy × 5 fractions (2 plans), 5 Gy × 5 fractions (2 plans), 24 Gy × 1 fraction (2 plans), 18 Gy × 1 fraction (2 plans), 12 Gy × 1 fraction (2 plans)
Arc numbers	3 full arcs (3 plans), 3 full arcs + 2 partial arcs (3 plans), 5 partial arcs (2 plans), 3 full arcs + 2 partial arcs (2 plans)
Target numbers	2 targets (2 plans), 1 target (8 plans)

**Table 2 life-11-01159-t002:** The collimator filed size and PTV size of plan validation.

Plan	Field Size (cm)		PTV Geometry	
	X-Jaws	Y-Jaws	Volume (cm^3^)	Equivalent Sphere Diameter (cm)
Plan No.1	5.28 ± 0.236	5.5 ± 0	30.31	3.9
Plan No.2	4.46 ± 0.115	5 ± 0.289	15.55	3.1
Plan No.3	2.92 ± 0.084	3.02 ± 0.045	4.46	2
Plan No.4	3.8 ± 0.082	4.1 ± 0.141	15.08	3.1
Plan No.5 *	2.58 ± 0.206	2.55 ± 0.238	1.33, 2.78	1.4, 1.7
Plan No.6	4.3 ± 0	3.9 ± 0	13.86	3
Plan No.7	6 ± 0	5.8 ± 0	28.86	3.8
Plan No.8	4.9 ± 0.164	5.67 ± 0.418	17.9	3.2
Plan No.9	3.28 ± 0.096	3.25 ± 0.129	5.5	2.2
Plan No.10 *	4.15 ± 0.173	4 ± 0	19.72, 5.77	3.4, 2.2
Mean ± SD	4.06 ± 1.023	4.2 ± 1.124	13.43 ± 9.779	2.75 ± 0.823

* The patient was treated with two targets.

**Table 3 life-11-01159-t003:** The EPID signal in CU with varying SID and dose rates.

SID (cm)	CU in Different Dose Rates							
	1400 MU/min	1200 MU/min	1000 MU/min	800 MU/min	600 MU/min	400 MU/min	Average	SD
**120**	765.26	765.51	768.61	770.10	772.14	773.76	769.23	3.45
**130**	655.66	656.27	657.59	659.10	660.32	661.09	658.34	2.19
**140**	567.53	568.59	569.72	570.86	571.63	572.38	570.12	1.85
**150**	497.18	498.20	499.05	499.62	500.32	500.81	499.20	1.35
**160**	439.42	440.20	440.75	441.19	441.81	442.16	440.92	1.20
**170**	391.38	391.92	392.30	392.81	393.18	393.53	392.52	0.81
**180**	351.14	351.52	351.89	352.17	352.26	352.85	352.02	0.64

**Table 4 life-11-01159-t004:** Results of GPR of our EPID-based dosimetry and MapCHECK measurements.

Plan	GPR (Mean ± SD)			
	EPID-Based Dosimetry		MapCHECK	
	3%/3 mm	2%/2 mm	3%/3 mm	2%/2 mm
Plan No.1	99.50 ± 0.34	97.38 ± 2.77	99.12 ± 1.85	96.45 ± 2.55
Plan No.2	99.32 ± 0.95	99.47 ± 3.12	98.85 ± 1.15	97.88 ± 2.39
Plan No.3	99.47 ± 0.63	98.36 ± 1.39	96.90 ± 1.57	94.83 ± 3.03
Plan No.4	99.19 ± 0.52	95.33 ± 2.95	98.75 ± 1.71	98.17 ± 2.63
Plan No.5	99.05 ± 0.63	98.42 ± 2.67	99.66 ± 2.1	98.61 ± 2.49
Plan No.6	99.8 ± 0.39	98.67 ± 1.86	99.42 ± 1.43	95.60 ± 3.37
Plan No.7	99.57 ± 0.59	98.24 ± 2.30	98.67 ± 2.31	96.37 ± 3.89
Plan No.8	99.77 ± 0.5	99.19 ± 2.69	97.64 ± 1.88	95.50 ± 3.91
Plan No.9	98.99 ± 0.73	95.55 ± 3.42	97.99 ± 2.14	96.45 ± 2.41
Plan No.10	99.18 ± 0.72	97.56 ± 2.65	98.67 ± 1.34	98.51 ± 2.75

## Data Availability

The data presented in this study are available on request from the corresponding author.

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
