# Peer review of "Planar EPID-Based Dosimetry for SRS and SRT Patient-Specific QA"

_life, 2021, doi:10.3390/life11111159_

Round 1
Reviewer 1 Report
The authors have developed an EPID image based dosimetry model for estimating dose delivered at depth. This is quite interesting to this reader as it addresses a very specific concern with FFF based SBRT. Their methods are reasonable and they comment on relevant factors for consideration when generating such a system. Their model fits are very accurate using this technique.
My specific comments are as follows:
Figure 3 is redundant and not helpful given Table 2.
Figure 5 should be reduced to just 5b (5a is redundant)
The authors need to include some geometry of the tumors treated in the validation plans. The collimator field sizes are quite large for SRS/SBRT.
They should comment on accuracy when considering smaller targets/fields. surely they must have some targets <2cm.
Author Response
Thank you for inviting us to submit a revised draft of our manuscript entitled, " Planar EPID-based dosimetry for SRS and SRT patient-specific QA" to MDPI journal. We also appreciate the time and effort you and each of the reviewers have dedicated to providing insightful feedback on ways to strengthen our paper. Thus, it is with great pleasure that we resubmit our article for further consideration. We have incorporated changes that reflect the detailed suggestions you have graciously provided. We also hope that our edits and the responses we provide below satisfactorily address all the issues and concerns you and the reviewers have noted.
To facilitate your review of our revisions, the following is a point-by-point response to the questions and comments delivered in the letter.

Reviewer 2 Report
The article is well written, but some minor improvements can be done in order to achieve a better presentation of the M&M and results section.
Author Response

(The authors gave the same response as above.)

Reviewer 3 Report
Dear All,
this manuscript is interesting, the study is well done.
The purpose was to show for SRS and SRT treatment the possibility to use an EPID-based dosimetry model to perform patient specific QA.
They developed for this, correction parameters to convert EPID image to absorbed dose at depth 10 cm.
The correction parameters were the dose response linearity with MU, beam profile correction, the collimator scatter and water kernel.
To validate the study the authors used a MapCHECK detector as gold standard and test on 10 SRS/SRT patients’ plans.
One of the aim of this study for the reader is to consider the possibility to use only the EPID (photon fluence and converted absorbed dose at water) detector for QA.
The problems of this study are the gold standard detector and the conclusion:
1/ Indeed the MapCHECK is not very accurate (for a small field of 2.5 x 2.5 cm² there will be only 9 diode control points!). It will be better for this study to use a SRS mapCHECK with more concentrated diodes in small area. Another solution is to use EBT films.
2/ For the conclusion the authors could not conclude on only 10 SRS/SRT plans that his model is validate. It is not sufficient and they should moderate the conclusion or introduce more plan control to obtain the collimator field size limit. For example for an IMRT plan with collimator field size less than 2.5 x 2.5 cm² should the EPID-based dosimetry model works ?
Questions:
Did you try to use your model without resetting the gantry to 0 degree and using the on-bord system on the MapCHECK accelerator?
Did you try to position PMMA on the table to have an equivalent of 10 cm of water and modify your model considering it?
Corrections in the abstract :
- at line 19 "... scatter kernel." should be "... water kernel.",
- last line (27) "... in-hose model." should be "... in-house model."
Author Response

(The authors gave the same response as above.)

Round 2
Reviewer 1 Report
The authors have addressed my main concerns.